# Comparison of Functional Outcomes after Anterior Cruciate Ligament Reconstruction with Meniscal Repair for Unstable versus Stable Meniscal Tears

**DOI:** 10.3390/diagnostics14090871

**Published:** 2024-04-23

**Authors:** Jin Hyuck Lee, Gyu Bin Lee, WooYong Chung, Ji Won Wang, Sun Gyu Han, Hye Chang Rhim, Seung-Beom Han, Ki-Mo Jang

**Affiliations:** 1Department of Sports Medical Center, Anam Hospital, Korea University College of Medicine, Seoul 02841, Republic of Korea; gnkfccc@hanmail.net (J.H.L.);; 2Department of Physical Medicine and Rehabilitation, Harvard Medical School/Spaulding Rehabilitation Hospital, Charlestown, MA 02129, USA; 3Foot & Ankle Research and Innovation Lab (FARIL), Department of Orthopaedic Surgery, Harvard Medical School, Massachusetts General Hospital, Boston MA 02114, USA; 4Department of Orthopaedic Surgery, Anam Hospital, Korea University College of Medicine, Seoul 02841, Republic of Korea; oshan@korea.ac.kr

**Keywords:** anterior cruciate ligament reconstruction, meniscal repair, muscle strength, dynamic postural stability, overall stability index

## Abstract

This study aimed to compare functional outcomes including knee muscle strength in the quadriceps and hamstrings, and proprioception, assessed through dynamic postural stability (overall stability index [OSI]) and self-reported outcomes in the operated and non-operated knees between anterior cruciate ligament reconstruction (ACLR) with meniscal repair for unstable (root and radial tears) and stable (longitudinal, horizontal, and bucket handle tears) meniscal tears. A total of 76 patients were randomly selected (41 with ACLR with meniscal repair for unstable meniscal tears and 35 with ACLR with meniscal repair for stable meniscal tears) at three different time points (preoperative, 6 months, and 12 months). Repeated measures analysis of variance was used to investigate the differences in outcomes for between-subject and within-subject factors. In the operated knees, there were no significant differences for functional outcomes between the two groups (all *p* > 0.05). In the non-operated knees, a significant difference was observed for the OSI between the two groups, which was significantly higher in ACLR with meniscal repair for unstable meniscal tears than for stable meniscal tears at 6 months (*p* < 0.001). Multiple linear regression analysis showed that age (*p* = 0.027), preoperative OSI in the operated knees (*p* = 0.005), and postoperative OSI in the operated knees at 6 months (*p* = 0.002) were significant and independent predictors for OSI in the non-operated knees at 6 months postoperatively. Therefore, while no differences were observed in functional outcomes between the two groups in the operated knees, dynamic postural stability was poorer at 6 months postoperatively in the non-operated knees of patients with ACLR with meniscal repair for unstable meniscal tears. Furthermore, a significant correlation was observed between preoperative/postoperative dynamic postural stability in the operated knees and postoperative dynamic postural stability in the non-operated knees. Hence, we recommend incorporating balance exercises for both knees in post-surgical rehabilitation, particularly for patients with unstable meniscal tears.

## 1. Introduction

Anterior cruciate ligament (ACL) reconstruction and meniscal repair are common surgeries requiring extensive rehabilitation to restore knee function, including range of motion (ROM) and weight-bearing (WB) capabilities. There are proprioceptive mechanoreceptors within the ACL and meniscus that can detect proprioceptive information. Damage to these receptors can reduce proprioception by altering somatosensory input to the central nervous system [1]. In particular, proprioception, the body’s ability to sense motion and position, can be impaired after knee injuries and surgery [2,3], resulting in difficulty stabilizing the knee joint. Recovering functional outcomes such as knee muscle performance and proprioception post-knee surgery, including ACL reconstruction (ACLR) and meniscal repair, is crucial for patients returning to daily activities and sports [4,5,6,7]. Insufficient recovery can lead to discomfort, reinjury, and dissatisfaction [8,9]. Therefore, rehabilitation treatment after knee surgery is essential for restoring knee function [10,11].

The menisci play an important role in knee biomechanical factors such as lubrication, joint shock absorption, joint congruence, and proprioception [12,13]. Hence, meniscus injury may cause various problems in the knee joint. In particular, traumatic meniscal injuries are often associated with ACL rupture [13,14] and consist of stable meniscal tears, including longitudinal, horizontal, and bucket handle tears and unstable meniscal tears, including root and radial tears [15]. Therefore, previous study has shown varied recovery trajectories based on the type of meniscal tear, highlighting the need for tailored rehabilitation approaches [15]. Furthermore, in several previous studies [15,16,17], there was a difference in rehabilitation programs, including ROM, WB, and muscle strengthening after surgery in patients with stable and unstable meniscal tears; in particular, restricted rehabilitation was recommended after meniscal repair for unstable meniscal tears [16,17,18]. Differences in rehabilitation methods, especially restricted rehabilitation, may affect the recovery of functional outcomes after surgery. Furthermore, despite the known impact of meniscal tear stability on ACL rupture recovery, a gap remains in comparative analyses of functional outcomes post-repair, a gap this study seeks to fill. Therefore, understanding how stable and unstable meniscal tears affect postoperative recovery is essential for optimizing rehabilitation strategies and enhancing patient outcomes.

This study aims to directly compare functional outcomes, focusing on knee muscle strength, proprioception, and self-reported satisfaction, between patients treated for stable versus unstable meniscal tears post-ACLR. We hypothesize that patients with stable meniscal repairs will exhibit superior functional outcomes.

## 2. Materials and Methods

### 2.1. Study Design and Patient Enrollment

The Institutional Review Board (2018AN0261) approved this prospective comparative study, and all procedures followed were in accordance with the ethical standards of the responsible committee on human experimentation and with the Helsinki Declaration. Informed consent was obtained from all patients and/or their legal guardians, and patient data were anonymized and securely stored to ensure confidentiality. A total of 330 patients who underwent primary ACLR with hamstring autografts or tibialis allografts between August 2018 and December 2021 were enrolled. Among these, 254 patients were excluded for the following reasons: (1) isolated ACLR; (2) bilateral injuries and revision ACLR; (3) other concomitant injuries [i.e., collateral ligament injuries, posterior cruciate ligament injuries, fractures, chronic meniscal tears (occurring beyond 12 weeks from the time of injury), and meniscectomy]; (4) Kellgren–Lawrence grade > 2; (5) vestibular and visual impairment and neurological pathology; (6) lost to follow-up. These patients were excluded for reasons that might confound the assessment of functional outcomes post-ACLR, including isolated ACLR and concomitant injuries. This selection aimed to isolate the impact of meniscal tear stability on recovery. The final analysis was based on data obtained from 76 patients (41 with ACLR with meniscal repair for unstable meniscal tears and 35 with ACLR with meniscal repair for stable meniscal tears) at three time points (preoperative, 6 months postoperative, and 12 months postoperative).

### 2.2. Rehabilitation Protocol

Our rehabilitation protocols, differentiated by meniscal tear stability, were designed based on evidence suggesting varied recovery trajectories [15,16,17,18,19]. Each protocol aims to optimize recovery while mitigating risk of reinjury. Participants engaged in the rehabilitation program once or twice per week for a minimum of 12 weeks. The protocols were categorized into two types based on stable and unstable meniscal tears. A comprehensive rehabilitation protocol is detailed in the Appendix A.

#### 2.2.1. Rehabilitation for Unstable Meniscal Tears

In patients who underwent ACLR with meniscal repair for unstable meniscal tears, ROM and WB, including squats and specific training, were restricted. The knee flexion ROM was at 90° at four weeks and full ROM at eight weeks. WB was allowed gradually, as non-WB before 6 weeks, tolerated partial WB at 6–8 weeks, and full WB without crutches after 8 weeks. Isometric open-kinetic chain (OKC, defined as exercises that are lower limb activities performed where the distal segment of the limb is free to move) exercise was started at 3 and 7 weeks for the quadriceps and hamstring muscles, respectively. General strengthening OKC training started with an elastic band at 6 weeks, and closed-kinetic chain (CKC, defined as exercises that are lower limb activities performed where the distal segment of the limb is fixed and stabilized) exercises started at 8 weeks. Stationary bicycling was started at 8 weeks. However, OKC exercises for the hamstring muscle were restricted to 10 weeks. In particular, the CKC squat exercise was performed in maximum 60° knee flexion until 16 weeks and in 90° knee flexion after 16 weeks. Single-leg balance exercises to improve proprioception were performed at 8 weeks, and running was initiated at 20 weeks postoperatively.

#### 2.2.2. Rehabilitation for Stable Meniscal Tears

In patients who underwent ACLR with meniscal repair for stable meniscal tears, the knee flexion ROM was 90° at 2 weeks and full ROM at 6 weeks. WB was allowed gradually, as partial WB ≤ 2 weeks, tolerated WB with crutch at 2–6 weeks, and full WB without crutches after 6 weeks. Isometric OKC exercises for the quadriceps and hamstring muscles were initiated at 3 weeks. Stationary bicycling and general strengthening training were started at 6 weeks. Squat exercises were performed gradually up to 90° from 10 weeks. Single-leg balance exercises to improve proprioception were performed at 6 weeks, and running was initiated at 16 weeks postoperatively.

Specific training for returning to sports was allowed at 36 weeks postoperatively for unstable meniscal tears and at 24 weeks postoperatively for stable meniscal tears. Core- and hip-strengthening exercises were performed equally in both groups. The major rehabilitation difference between the two groups was that weight bearing, knee flexion ROM, and functional training such as running was limited in the unstable meniscal tear group.

### 2.3. Outcome Measures

Outcome measures were chosen to encompass a broad range of functional outcomes, from muscle strength and proprioception to self-reported outcomes, aligning with our study aim to comprehensively assess recovery post-ACLR.

#### 2.3.1. Knee Muscle Strength

The knee muscle strength of the quadriceps and hamstring muscles was assessed using an isokinetic device (Biodex Multi-Joint System 4, Biodex Medical Systems, Inc., Shirley, NY, USA). All participants sat on an isokinetic dynamometer chair and performed five repetitions of knee flexion and extension at 60°/s for each leg, and the maximal peak torque was recorded (Figure 1). Muscle strength was recorded after normalizing peak torque to body weight (Nm kg^−1^ × 100) [20].

#### 2.3.2. Proprioception 

Proprioception was evaluated through dynamic postural stability using the Biodex Stability System (BSS; Biodex Medical Systems, Shirley, NY, USA), and the results were recorded as the overall stability index (OSI) [21]. The foot platform of the BSS can be tilted from 0° to 20° with a 360° rotation, and the stability level automatically decreases from level 12 (most stable) to level 1 (most unreliable) by one every 1.66 s. A dynamic postural stability test was conducted with the patient standing barefoot on one leg. The patients completed two tests of 20 s each, with 10 s of rest between the tests. The mean and standard deviation of the two trials was calculated using the stability system. A higher index indicated poorer dynamic postural stability [2].

#### 2.3.3. Self-Reported Outcomes

Self-reported outcomes included the Lysholm and International Knee Documentation Committee (IKDC) scores [22,23]. The Lysholm score is maximum 100 points and consists of eight items (limping, support, restraining, instability, pain, swelling, climbing stairs, and squatting). Lysholm scores of ≥91, 84–90, 65–83, and ≤64 points indicate excellent, good, fair, and poor outcomes, respectively [22]. The IKDC score has a maximum of 100 points and consists of three items (symptoms, function, and sports activities) [24]. The IKDC is a validated assessment tool for ACLR with concomitant meniscal repair [25,26]. Higher Lysholm and IKDC scores indicate less disability and fewer symptoms. 

### 2.4. Statistical Analysis

Based on a previous study [27] of OSI for postural stability in patients with ACLR to determine the sample size, an intergroup difference in OSI > 0.5 was considered clinically significant. An a priori power analysis (α = 0.05, power = 0.8) was calculated using repeated measures analysis of variance (RM-ANOVA), and a minimum of 21 patients (effect size f(V): 0.613, *p* (η2) = 0.144) in each group would be required to detect an OSI difference > 0.5 between the groups (α = 0.05, power = 0.8). In this study, the power to detect a clinically significant difference between the groups was 0.815. The effect size for our power analysis was chosen based on clinical significance derived from prior research, ensuring our study is adequately powered to detect meaningful differences in OSI between groups.

Statistical methods, including an independent *t*-test and chi-square test, were selected based on their appropriateness for analyzing continuous outcomes and categorical variables, respectively. Assumptions of normal distribution and equal variance were verified to ensure the validity of our findings. RM-ANOVA was used to investigate the differences in outcomes for between-subject factors (groups) and within-subject factors (time: preoperative, 6 months, and 12 months). A Tukey-HSD post-hoc test was applied if a significant interaction between group-by-time was found and corrected for *p*-value < 0.017. To determine the effect size, partial eta squared (η2) was used, defined as small for <0.06, as medium for 0.06 < x < 0.14, and as large for >0.14 [28]. Multiple linear regression analysis was used to identify the influence of predictor variables on the dependent variable (postoperative OSI in non-operated knees at 6 months) in operated knees. Statistical analyses were performed using IBM^®^ SPSS^®^ Statistics 20 (SPSS Inc., Chicago, IL, USA), and statistical significance was set at *p* < 0.05.

## 3. Results

In summary, there were no significant differences in functional outcomes between the two groups in the operated knees. However, dynamic postural stability was poorer in ACLR with meniscal repair for unstable meniscal tears at 6 months postoperatively in the non-operated knees. Correlation analysis revealed that age and preoperative/postoperative OSI in the operated knees were significant predictors of postoperative OSI in the non-operated knees.

### 3.1. Demographic Data

There were no statistically significant differences in demographic data relevant to the study’s objectives between the two groups (*p* > 0.05, Table 1).

### 3.2. Comparison of Functional Outcomes between the Two Groups

In the operated knees, no statistically significant differences were found in quadriceps and hamstring strength, OSI, and Lysholm and IKDC scores (all *p* > 0.05) (Table 2 and Table 3), indicating that there was no difference in functional outcomes between the two groups.

In the non-operated knees, no statistically significant differences were found in quadriceps and hamstring strength and Lysholm and IKDC scores (all *p* > 0.05) (Table 2 and Table 3). However, a significant group effect was found for the OSI with a large effect size (F = 8.912, *p* = 0.004, η^2^ = 0.107), indicating that dynamic postural stability significantly differed between groups. In addition, a significant group-by-time interaction effect was identified for the OSI with a medium effect size (F = 5.516, *p* = 0.007, η^2^ = 0.069). The post-hoc tests showed that at postoperative 6 months (95% CI = 0.5 to 1.2, *p* < 0.001), OSI was significantly higher in ACLR with meniscal repair for unstable meniscal tears compared with stable meniscal tears (Figure 2), indicating that dynamic postural stability was poorer in ACLR with meniscal repair for unstable meniscal tears compared with ACLR with meniscal repair for stable meniscal tears.

### 3.3. Correlation and Predictor Factors

Only the OSI of the non-operated knees differed significantly at 6 months postoperatively between the two groups; thus, correlation analyses were performed between various parameters and the postoperative OSI of the non-operated knees. Univariate analysis showed that age (r = −0.260, *p* = 0.023), site (medial meniscus [MM] or lateral meniscus [LM]) of meniscal tear (r = −0.329, *p* = 0.004), preoperative OSI in the operated knees (r = 0.752, *p* < 0.001), and postoperative OSI in the operated knees at 6 months (r = 0.528, *p* < 0.001) were significantly correlated with the postoperative OSI score in the non-operated knees at 6 months (Table 4). Multiple linear regression analysis of these four parameters showed that age (β = −0.206, *p* = 0.027), preoperative OSI in the operated knees (β = 0.313, *p* = 0.005), and postoperative OSI in the operated knees at 6 months (β = 0.358, *p* = 0.002) were significant and independent predictors of postoperative OSI in the non-operated knees at 6 months (Table 4). Therefore, identifying age and preoperative/postoperative OSI as significant predictors may be important for preoperative counseling and postoperative rehabilitation planning.

## 4. Discussion 

This study aimed to directly compare the functional outcomes such as knee muscle strength, proprioception, and self-reported outcomes between patients undergoing ACLR with meniscal repair for stable and unstable meniscal tears. The main finding of the present study was that there were no significant differences in functional outcomes in the operated knees between ACLR with meniscal repair for stable and unstable meniscal tears. The lack of difference in functional outcomes between stable and unstable tears challenges the prevailing assumption that functional outcomes would be better in patients undergoing ACLR with meniscal repair for stable meniscal tears, suggesting that the meniscal tear type in ACLR may not affect the functional outcomes in the operated knees. However, in the non-operated knees, dynamic postural stability was poorer in ACLR with meniscal repair for unstable meniscal tears than in ACLR with meniscal repair for stable meniscal tears at 6 months postoperatively. Furthermore, age and preoperative/postoperative OSI in the operated knees were significant predictors of postoperative OSI in the non-operated knees.

In this study, there were no differences in quadriceps and hamstring strengths between ACLR with meniscal repair for stable and unstable meniscal tears. Although the reason for the results of this study is unclear, it may be explained by the training method used to improve knee muscle strength. In the present study, OKC exercises for the quadriceps muscle were initiated during the same time period in both groups, except for CKC exercises including WB. Several previous studies reported that OKC exercise is effective in improving concentric quadriceps [29,30] and hamstring [30] muscle strength. Similarly, Morrissey et al. [31] investigated isokinetic knee muscle strength in 36 patients with ACLR and found no difference in knee muscle strength between OKC and CKC training, indicating that OKC is as effective in improving knee muscle strength as CKC [32]. The testing mode for muscle strength evaluation in our study also was performed in the concentric-contraction mode, and the concentric-contraction mode is the same motion as that used in OKC. Therefore, future study should explore this possibility further.

In this study, there was no difference in the dynamic postural stability in the operated knees between ACLR with meniscal repair for stable and unstable meniscal tears. A previous study has shown that dynamic postural stability was poorer in ACL rupture with MM tears than in isolated ACL rupture [27], indicating that meniscus injury may affect dynamic postural stability. In addition, Lee et al. investigated dynamic postural stability in 93 patients with ACL rupture with MM and LM tears [33] and found that dynamic postural stability was worse in patients with ACL rupture with LM tears than in those with ACL rupture with MM tears. Within the meniscus, mechanoreceptors exist as proprioceptors [3,12], which detect the position and movement of the knee joint [34]; hence, meniscus injuries may affect postural stability [27,33]. In particular, there are more mechanoreceptors for detecting dynamic postural stability in the LM than in the MM [35,36]; hence, dynamic postural stability may decrease more with LM injury [33]. Our results showed no difference in the rate of MM/LM tears between groups. Therefore, there may be no difference in dynamic postural stability between the groups; that is, whether the presence or absence of MM or LM meniscal tear may be more important for dynamic postural stability than the type of meniscal tear. 

Interestingly, in non-operated knees, dynamic postural stability was poorer after ACLR with meniscal repair for unstable meniscal tears than after ACLR with meniscal repair for stable meniscal tears at 6 months postoperatively. This suggests that altered central somatosensory pathways may contribute to dynamic postural stability. Valeriani et al. found that central somatosensory pathways may be functionally altered after ACL injury [37], and sensory input from altered central somatosensory pathways may affect both lower extremities [34]. The lower extremities are used synchronously in physical activities, including walking and standing; thus, biomechanical stress on one side can naturally increase the use of the other. In this study, in unstable meniscal tears, full WB after knee surgery was restricted for the operated knees for 8 weeks, which may result in a decrease in dynamic postural stability due to increased stress due to excessive load on the non-operated knees. However, since the same activities were performed regardless of the meniscal tear types after 6 months postoperatively, there may be no difference in dynamic postural stability at 1 year postoperatively. Furthermore, we found that age was a predictor of postoperative OSI in non-operated knees at 6 months, indicating that as age increases, postural stability may worsen [38]. Especially, we also found that the preoperative OSI of the operated knees and postoperative OSI of the operated knees at 6 months were significant predictors of the postoperative OSI of the non-operated knees at 6 months, indicating bilateral impairment of proprioception after ACL rupture [21,39] or meniscal tear [40]. Altered afferent input from injured knees may influence motor responses in intact knees owing to cross-connections from the cerebral cortex [39]. Therefore, the identification of age and preoperative/postoperative OSI as significant predictors highlights the role of preoperative/postoperative balancing exercises in both operated and non-operated knees, suggesting that they should be considered importantly for preoperative counseling and rehabilitation planning.

Our results showed no differences in self-reported outcomes between ACLR with meniscal repair for stable and unstable meniscal tears. This finding indicates that meniscal tears in patients who underwent ACLR may not be significantly associated with self-reported outcomes. Several recent studies reported that self-reported outcome scores after ACLR were not influenced by meniscal treatment [4,41]. However, previous studies reported that psychological outcomes (kinesiophobia) may be an important factor in evaluating functional recovery after ACL and meniscal injuries [5,6,42]. Therefore, future study should investigate the impact of psychological outcomes such as kinesiophobia on recovery of functional outcomes using screening tools such as the Tampa scale for kinesiophobia. This may provide valuable insight into how psychological factors are related to functional outcomes after ACLR.

This study had several limitations. Firstly, it lacked a normal control group. The absence of a normal control group limits our ability to compare our findings with baseline levels of functional outcomes. Hence, future studies should include a comparison of functional recovery with a normal control group post-surgery. Secondly, we could not control for participants’ outside activities. Controlling for outside activities could provide a clearer effect of rehabilitation. Thirdly, functional performance evaluations, such as hops and vertical jumps, were not conducted. Radial or root meniscal tears, susceptible to axial loads due to the loss of hoop stress, may indicate potential differences in functional performance between stable and unstable meniscal tears [12,43], indicating possible differences in functional performance between stable and unstable meniscal tears. Therefore, by incorporating evaluations of hops and vertical jumps, future studies could more comprehensively assess the impact of meniscal tear stability on functional performance, potentially revealing the impact of meniscal tears on functional outcomes in patients with ACL ruptures. Consequently, high-quality randomized controlled trials with large sample sizes are imperative to validate this study’s findings. Nevertheless, this is the first study to directly compare the functional outcomes such as knee muscle strength, proprioception, and self-reported outcomes between patients undergoing ACLR with meniscal repair for stable and unstable meniscal tears.

### Clinical Implication

A restricted rehabilitation protocol after ACLR with meniscal repair for unstable meniscal tears might not reduce knee function in the operated knees after surgery [4]. Furthermore, our findings have direct clinical implications, suggesting that clinicians should consider the stability of non-operated knees in designing rehabilitation protocols after ACLR, especially for patients with unstable meniscal tears. While we recommend balance exercises for both knees to enhance postoperative recovery, especially in patients with unstable meniscal tears, it can be crucial to tailor these exercises to individual patient needs and consider other factors such as age and overall physical condition.

## 5. Conclusions

Our study found no significant differences in functional outcomes such as knee muscle strength, dynamic postural stability, and self-reported outcomes between ACLR with meniscal repair for stable versus unstable meniscal tears in the operated knees. However, a novel finding of our study is the poorer dynamic postural stability observed at 6 months postoperatively in the non-operated knees for patients with unstable meniscal tears, a crucial insight for postoperative rehabilitation. Furthermore, given the observed correlation between preoperative/postoperative dynamic postural stability in the operated knees and postoperative dynamic postural stability in the non-operated knees, we recommend incorporating balance exercises for both knees in postoperative rehabilitation, particularly for patients with unstable meniscal tears. Further study is needed to explore the long-term effects of different rehabilitation protocols on dynamic postural stability in non-operated knees, as well as the potential biomechanical and neurological mechanisms underpinning our findings.

## Figures and Tables

**Figure 1 diagnostics-14-00871-f001:**
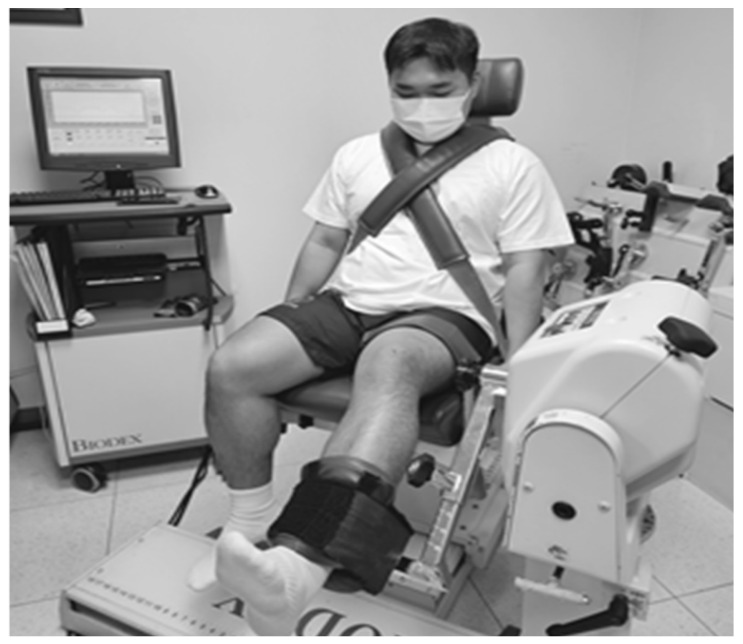
Isokinetic dynamometer test position.

**Figure 2 diagnostics-14-00871-f002:**
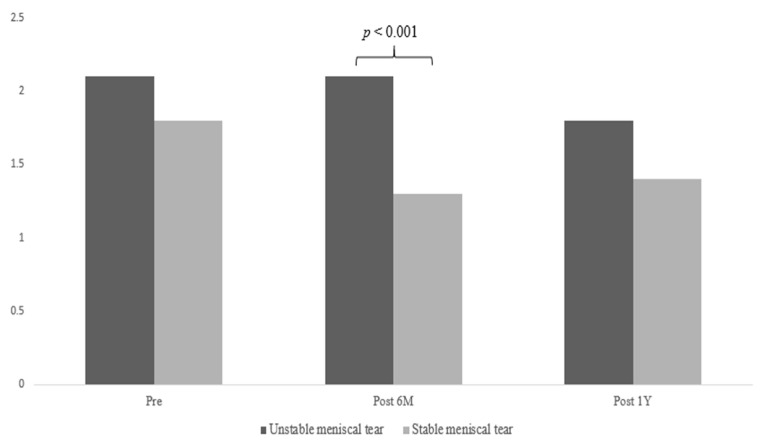
Comparison of dynamic postural stability on the non-operated knees between the two groups. OSI: overall stability.

**Table 1 diagnostics-14-00871-t001:** Demographic data of study patients by group.

	Unstable Meniscal Tear (n = 41)	Stable Meniscal Tear (n = 35)	*p*-Value
Sex (male/female)	28/13	23/12	0.577
Age (years) ^a^	33.2 ± 11.7	35.5 ± 10.5	0.420
Height (cm) ^a^	171.4 ± 7.4	171.6 ± 7.9	0.860
Weight (kg) ^a^	73.6 ± 12.6	74.7 ± 12.9	0.538
Body mass index (kg/m^2^) ^a^	24.9 ± 3.7	25.4 ± 3.8	0.546
Site of meniscal tear (MM/LM)	13/27	16/19	0.407
Injured side (right/left)	26/15	22/13	0.921
Dominant knee (right/left)	26/15	20/15	0.715
Sports and activity, n (Low:High) ^a^	10/21	15/18	0.126

^a^ The values are expressed as mean ± standard deviation, which means the average value plus and minus the variability around this average.

**Table 2 diagnostics-14-00871-t002:** Comparison of knee muscle strength between the two groups.

		Unstable Meniscal Tear (n = 23)	Stable Meniscal Tear (n = 35)				
Variable	Time	M ± SD	M ± SD	Source	F	P^1^(η^2^)	P^2^
Quadriceps strength (operated knees)	Pre Post 6M Post 12M	112.3 ± 39.9 154.1 ± 46.1 184.9 ± 52.7	127.4 ± 63.6 149.0 ± 58.0 185.5 ± 63.0	Group Time Group * Time	0.095 88.268 2.261	0.759 (0.001)	
						<0.001 (0.544)	-
						0.113 (0.030)	
Quadriceps strength (non-operated knees)	Pre Post 6M Post 12M	204.9 ± 61.9 224.7 ± 57.7 225.6 ± 60.7	189.8 ± 57.8 224.3 ± 42.3 236.4 ± 54.8	Group Time Group * Time	0.019 19.913 2.626	0.891 (0.001)	
						<0.001 (0.212)	-
						0.082 (0.034)	
Hamstring strength (operated knees)	Pre Post 6M Post 12M	62.2 ± 29.5 89.3 ± 30.7 101.3 ± 28.9	61.6 ± 28.3 87.6 ± 28.4 106.9 ± 35.9	Group Time Group * Time	0.036 82.840 0.717	0.850 (0.001)	
						<0.001 (0.528)	-
						0.490 (0.010)	
Hamstring strength (non-operated knees)	Pre Post 6M Post 12M	91.9 ± 32.9 108.4 ± 31.6 112.2 ± 33.1	94.2 ± 27.4 109.3 ± 29.0 116.6 ± 30.1	Group Time Group * Time	0.153 31.643 0.200	0.697 (0.002)	
						<0.001 (0.300)	-
						0.798 (0.003)	

M, mean; SD, standard deviation; P^1^, RM-ANOVA; P^2^, post-hoc; η^2^, effect size (partial eta squared). I = preoperative. II = postoperative 6M. III = postoperative 12M. Measurement unit of muscle strength was Newton meter per kilogram.

**Table 3 diagnostics-14-00871-t003:** Comparison of dynamic postural stability and self-reported outcomes between the two groups.

		Unstable Meniscal Tear (n = 23)	Stable Meniscal Tear (n = 35)				
Variable	Time	M ± SD	M ± SD	Source	F	P^1^(η^2^)	P^2^
OSI (operated knees)	Pre Post 6M Post 12M	2.3 ± 1.2 2.0 ± 1.2 1.7 ± 0.8	2.0 ± 0.8 1.6 ± 0.8 1.5 ± 0.7	Group Time Group * Time	4.305 9.143 0.583	0.061 (0.055)	
						<0.001 (0.110)	-
						0.552 (0.008)	
OSI (non-operated knees)	Pre Post 6M Post 12	2.1 ± 1.3 2.1 ± 0.9 1.8 ± 0.9	1.8 ± 0.7 1.3 ± 0.4 1.4 ± 0.6	Group Time Group * Time	8.912 8.140 5.516	**0.004 (0.107)**	I = 0.131 **II < 0.001** III = 0.075
						0.001 (0.099)	
						**0.007 (0.069)**	
Lysholm score	Pre Post 6M Post 12M	54.8 ± 17.5 75.4 ± 14.0 82.4 ± 9.5	55.9 ± 19.0 71.3 ± 17.4 85.8 ± 12.7	Group Time Group * Time	0.581 103.441 1.166	0.448 (0.008)	
						<0.001 (0.583)	-
						0.313 (0.016)	
IKDC score	Pre Post 6M Post 12M	52.6 ± 13.5 65.1 ± 10.4 73.1 ± 13.3	50.2 ± 12.2 66.3 ± 11.9 73.7 ± 12.9	Group Time Group * Time	0.007 81.760 0.605	0.931 (0.001)	
						<0.001 (0.525)	-
						0.523 (0.008)	

OSI, overall stability index; IKDC, International Knee Documentation Committee; M, mean; SD, standard deviation; P^1^, RM-ANOVA; P^2^, post-hoc; η^2^, effect size (partial eta squared). I = preoperative. II = postoperative 6M. III = postoperative 12M. Measurement unit of OSI was degree. Measurement unit of Lysholm and IKDC scores was point.

**Table 4 diagnostics-14-00871-t004:** Multiple linear regression analysis on predictors for the postoperative OSI in the non-operated knees.

Dependent Variable	Independent Variables	Unstandardized Coefficients		Standardized Coefficients	
		B	SE(B)	β	*p*-Value
Postoperative OSI at 6 months (non-operated knees)	Age	−0.016	0.007	−0.206	**0.027 ***
	Preoperative OSI (operated knees)	0.260	0.089	0.313	**0.005 ***
	Postoperative OSI at 6 months (operated knees)	0.300	0.091	0.358	**0.002 ***

OSI, overall stability index. * *p* < 0.05.

## Data Availability

The data presented in this study are available on request from the corresponding author.

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
