# Peer review of "Comparison of Functional Outcomes after Anterior Cruciate Ligament Reconstruction with Meniscal Repair for Unstable versus Stable Meniscal Tears"

_diagnostics, 2024, doi:10.3390/diagnostics14090871_

Round 1
Reviewer 1 Report
Comments and Suggestions for Authors
1. My suggestion would be for the authors to develop more on the concept of proprioception in the context of knee injuries and surgery, in the introduction part
2. authors should define "functional outcomes" early in the introduction. which measures (specific muscle strength tests, proprioception tests, and PROMs) were used for this?
3. how did the authors qualify a meniscus tear as being chronic?
4. what were the specific protocols used, including exercise intensities or progression criteria?
5. please provide an explanation and reason for each of the statistical methods used
6. what was the strategy for adjusting for multiple comparisons/biases in all aspects of the analysis?
7. when discussing limitations, provide a detailed analysis of how it might affect the results and conclusions of the study
Reviewer 2 Report
Comments and Suggestions for Authors
Very interesting manuscript with a lot of data. The conclusions make it possible to optimize the physiotherapy to be applied so that the individual recovers good physical condition.
Before this manuscript is published, I think it should be revised on a few points that I'll emphasize throughout the text.
I would emphasize that the software used should be marked with a registered trademark (TM). Review some of the formatting and organization of the tables throughout the text.
In the description of the procedure, I think it would be interesting to add a diagram or image to enrich the whole work. The quality of the image in figure 1 should be reviewed.
Congratulations on your work.

Reviewer 3 Report
Comments and Suggestions for Authors
Here are my comments on the article, focusing on several minor aspects that could enhance its clarity and scientific rigour:
1. My recommendation is to find an alternative for the title that will make it shorter and easier to comprehend
2. Authors should explain their choice on IKDC and compared to other PROMS
3. The references are slightly outdated. I recommend refreshing their citations
4. Can the authors elaborate on their limitations impact on clinical applicability of these results?
5. Figure 1 quality should be improved. It is blurry and barely read-able.
6. Why was Tegner scale left out if PROMS?
Results are properly presented, methods clearly delineate step by step procedures.
Reviewer 4 Report
Comments and Suggestions for Authors
For Abstract
1. Clarity and Precision in Reporting Results: The text states, "In operated knees, no group or group-by-time interaction effect was found," but later mentions significant differences in the non-operated knees. It would be more clear to explicitly state at the beginning that the study examines both operated and non-operated knees, clarifying the hypothesis being tested for each.
Suggestion: Begin with a clearer overview of the study's dual focus on operated and non-operated knees, specifying the anticipated outcomes for each.
2. Methodological Detail: The study's methodology, particularly the criteria for classifying meniscal tears as stable or unstable, is not mentioned in the abstract. Including this information would strengthen the reader's understanding of the study's scope and the basis for comparisons.
Suggestion: Add a sentence explaining how meniscal tears were classified into stable and unstable categories.
3. Statistical Analysis Clarity: The mention of a "significant group-by-time interaction effect (F = 5.516)" is presented without sufficient context regarding the statistical model used, making it challenging for readers to evaluate the study's analytical rigor.
Suggestion: Briefly describe the statistical model employed for analysis, ensuring readers can understand the framework for interpreting results.
4. Significance and Implications of Findings: While the abstract notes differences in dynamic postural stability, it does not adequately explore the clinical or practical significance of these findings. A discussion about why these differences matter would be beneficial.
Suggestion: Include a sentence or two discussing the potential clinical implications of poorer dynamic postural stability in non-operated knees, such as increased risk of injury or implications for rehabilitation strategies.
5. Predictors of Postural Stability: The abstract identifies age and pre-/post-operative OSI as significant predictors without discussing the implications of these predictors or how they might influence treatment or prognosis.
Suggestion: Expand on the significance of these predictors in the context of ACLR and meniscal repair outcomes, perhaps suggesting how they could guide clinical decisions or patient counseling.
6. Language and Grammar: The abstract exhibits minor grammatical issues and instances of awkward phrasing ("for quadriceps and hamstring muscles, proprioception through dynamic postural stability"), which could detract from its professional tone.
Suggestion: Revise for grammatical accuracy and fluidity, perhaps rephrasing to "including knee muscle strength in the quadriceps and hamstrings, and proprioception, assessed through dynamic postural stability..."
7. Keywords Consistency: Ensure that the keywords used at the end of the abstract precisely reflect the core elements of the study. For instance, if specific measures like the Lysholm score are crucial to your findings, they should be included in your keywords.
Suggestion: Review and possibly update the keywords to include all relevant terms that reflect the study's main focuses and outcomes.
For Introduction
1. Clear and Concise Background Information:
The text provides a general overview but could be more concise and focused. Shorten the background to include only the most directly relevant information.
Suggestion: "Recovering functional outcomes like knee muscle performance and proprioception post-knee surgery, including ACL reconstruction and meniscal repair, is crucial for patients returning to daily activities and sports. Insufficient recovery can lead to discomfort, reinjury, and dissatisfaction."
2. Specificity in Citing Previous Studies:
The text mentions previous studies [1-6, 8-14] but lacks specificity about how these studies directly relate to the current research.
Suggestion: "Prior research [references] has shown varied recovery trajectories based on the type of meniscal tear, highlighting the need for tailored rehabilitation approaches."
3. Clarify the Importance of the Study:
The introduction implies the study's importance but does not explicitly state why understanding differences in functional outcomes based on tear stability matters.
Suggestion: "Understanding how stable and unstable meniscal tears affect post-surgery recovery is essential for optimizing rehabilitation strategies and enhancing patient outcomes."
4. Technical Terms and Acronyms:
While the introduction uses acronyms like ACL and ROM, it assumes a level of reader familiarity. Define all acronyms and technical terms upon first use.
Suggestion: "Anterior cruciate ligament (ACL) reconstruction and meniscal repair are common surgeries requiring extensive rehabilitation to restore knee function, including range of motion (ROM) and weight-bearing (WB) capabilities."
5. Objective and Hypothesis Clarity:
The objective and hypothesis are mentioned but could be stated more clearly and directly.
Suggestion: "This study aims to directly compare functional outcomes—focusing on knee muscle strength, proprioception, and self-reported satisfaction—between patients treated for stable versus unstable meniscal tears post-ACLR. We hypothesize that patients with stable meniscal repairs will exhibit superior functional outcomes."
6. Address the Gap in Research:
The text notes a lack of research but could better emphasize the significance of filling this gap.
Suggestion: "Despite the known impact of meniscal tear stability on ACL rupture recovery, a gap remains in comparative analyses of functional outcomes post-repair, a gap this study seeks to fill."
7. Improve Flow and Coherence:
The introduction's flow can be improved by better transitioning between ideas, making it easier for the reader to follow the narrative from the problem statement to the study's objectives.
Suggestion: Use transitional phrases and ensure each paragraph logically leads to the next, building a cohesive argument for the study's necessity.
For Materials and Methods:
1. Study Design and Patient Enrollment:
Detail on the selection criteria is good, but the rationale behind excluding certain patient groups could be elaborated. This would help clarify the study’s focus and the generalizability of its findings.
Suggestion: "Patients were excluded for reasons that might confound the assessment of functional outcomes post-ACLR, including isolated ACLR and concomitant injuries. This selection aimed to isolate the impact of meniscal tear stability on recovery."
2. Rehabilitation Protocol:
While the rehabilitation protocols are described in detail, a rationale for the chosen protocols based on previous studies could be better highlighted.
Suggestion: "Our rehabilitation protocols, differentiated by meniscal tear stability, were designed based on evidence suggesting varied recovery trajectories [11-15]. Each protocol aims to optimize recovery while mitigating risk of reinjury."
3. Outcome Measures:
The description of outcome measures is comprehensive but integrating how these measures relate to the study’s objectives could provide clearer insights into their selection.
Suggestion: "Outcome measures were chosen to encompass a broad range of functional outcomes, from muscle strength and proprioception to patient-reported satisfaction, aligning with our study’s aim to comprehensively assess recovery post-ACLR."
4. Statistical Analysis:
The statistical methods are well-detailed. However, explaining the choice of statistical tests and the assumptions behind them would strengthen this section. Additionally, discussing the handling of any potential data issues, such as missing data, would enhance transparency and rigor.
Suggestion: "Statistical methods, including RM-ANOVA and independent t-tests, were selected based on their appropriateness for analyzing continuous outcomes and categorical variables, respectively. Assumptions of normal distribution and equal variance were verified to ensure the validity of our findings."
5. Clarity and Conciseness:
The section is dense and contains a lot of information. Breaking it down into more digestible paragraphs and using subheadings for each major area (e.g., "2.2.1 Rehabilitation for Unstable Meniscal Tears") would improve readability.
Suggestion: Organize the rehabilitation protocol section with subheadings for each meniscal tear type and summarize key components in bullet points for clarity.
6. Technical Details and Definitions:
Some technical terms and procedures are mentioned without definitions or explanations, which could pose difficulties for readers not familiar with the field.
Suggestion: Define technical terms and acronyms the first time they are used and provide brief explanations of less common procedures or measures.
7. Ethical Considerations:
The mention of IRB approval and informed consent is appropriate. Additional details on ethical considerations, especially regarding patient confidentiality and the handling of patient data, would further strengthen this section.
Suggestion: "All procedures followed were in accordance with the ethical standards of the responsible committee on human experimentation and with the Helsinki Declaration. Patient data were anonymized and securely stored to ensure confidentiality."
8. Sample Size Justification and Power Analysis:
The text provides a good explanation of the power analysis. Further justification for the chosen effect size and the implications of the study's power for detecting clinically significant differences would be beneficial.
Suggestion: "The effect size for our power analysis was chosen based on clinical significance derived from prior research, ensuring our study is adequately powered to detect meaningful differences in OSI between groups."
For Results:
1. Clear Presentation of Demographic Data:
The presentation of demographic data is straightforward, but the significance of these demographics in relation to the study's aims could be clarified.
Suggestion: Briefly discuss the relevance of the demographic data in the context of the study's objectives. For instance, mention if the balance in demographics between groups supports the comparability of the cohorts.
2. Descriptive Statistics and Terminology:
The use of mean ± standard deviation is appropriate, but ensure that all readers can understand these statistical terms by briefly explaining them the first time they are used.
Suggestion: In the first instance of presenting data as mean ± standard deviation, a footnote or sentence could explain that this format shows the average value plus and minus the variability around this average.
3. Use of Tables and Figures:
The text references multiple tables and possibly figures but does not describe them in detail within the text. Integrating key findings from tables and figures into the narrative enhances readability and comprehension.
Suggestion: Summarize the most important or significant data from the tables/figures in the text, ensuring that readers can understand the findings without constantly referring to the tables.
4. Statistical Analysis Results:
The presentation of statistical findings is dense and could be streamlined for better readability. Moreover, explaining the implications of these findings in layman's terms could enhance understanding.
Suggestion: After presenting a statistical result, briefly explain its meaning. For example, "A significant group effect was found for the OSI, indicating that dynamic postural stability significantly differed between groups."
5. Effect Sizes and Statistical Significance:
The section includes effect sizes and p-values but could more explicitly link these to the practical significance of the findings.
Suggestion: Explain the practical implications of significant findings, such as how differences in OSI might affect patient recovery or rehabilitation strategies.
6. Multiple Linear Regression Analysis:
While the regression analysis is appropriately detailed, a clearer explanation of why these predictors are important and how they might be used in clinical practice could strengthen the section.
Suggestion: Discuss the clinical relevance of identifying these predictors. For example, understanding that preoperative OSI is a predictor could influence pre-surgical counseling and rehabilitation planning.
7. Consistency and Formatting:
Ensure consistency in formatting and presentation of statistical data across the text, tables, and figures. Consistent use of terminology and statistical notation is crucial.
Suggestion: Review the entire section for consistency in terminology (e.g., "postoperative" vs. "post-operated") and format (e.g., decimal places, use of parentheses for effect sizes).
8. Narrative Flow and Integration of Results:
The narrative flow between demographic data, functional outcomes, and regression analysis could be smoother to guide the reader through the findings logically.
Suggestion: Start with a summary statement that captures the overall findings before diving into specifics. This approach can help link the various parts of the Results section together coherently.
For Discussion:
1. Contextualizing Findings Within Existing Literature:
While the discussion mentions several studies, it could benefit from a more detailed comparison and contrast with these studies to highlight where findings align or diverge.
Suggestion: "Our findings are consistent with/in contrast to [Author]'s study, which found [findings], underscoring the importance of [context]. This difference/similarity suggests that [implication]."
2. Clarity in Explaining the Significance of Findings:
The discussion makes broad statements about the implications of the findings but could provide a clearer explanation of why these findings are significant for clinical practice or future research.
Suggestion: "The lack of difference in functional outcomes between stable and unstable meniscal tears challenges the prevailing assumption that [assumption], suggesting that [clinical implication]."
3. Addressing Unexpected or Inconclusive Findings:
The text mentions that the reason for some results is unclear. Providing hypotheses or potential explanations for these unexpected findings would strengthen the discussion.
Suggestion: "Although the reasons for [specific finding] remain unclear, one potential explanation could be [explanation]. Future research should explore this possibility further."
4. Expanding on Predictors of Postoperative OSI:
The discussion identifies significant predictors of postoperative OSI but could delve deeper into the mechanisms by which these predictors influence outcomes.
Suggestion: "The identification of age and pre/postoperative OSI as significant predictors highlights the role of [mechanism], suggesting that [clinical or research implication]."
5. Limitations:
While the discussion briefly mentions limitations, it could provide more detail on how these limitations impact the interpretation of the results and suggestions for how future research could address these limitations.
Suggestion: "The absence of a normal control group limits our ability to compare our findings with baseline levels of functional outcomes. Future studies should include such comparisons to [purpose of comparison]. Additionally, controlling for outside activities could provide insights into [specific insight]."
6. Future Research Directions:
The discussion could offer more specific recommendations for future research, including potential methodologies, populations, or variables of interest.
Suggestion: "Future research should investigate the impact of psychological factors like kinesiophobia on functional recovery using [methodological suggestion], as this could provide valuable insights into [specific insight]."
7. Integration of Functional Performance Evaluations:
The mention of not evaluating functional performances such as hops and vertical jumps could be linked to a recommendation for how these measures could provide additional insights.
Suggestion: "By incorporating evaluations of hops and vertical jumps, future studies could more comprehensively assess the impact of meniscal tear stability on functional performance, potentially revealing [specific insight]."
8. Technical Terms and Accessibility:
The discussion uses several technical terms that are well-explained but could be made more accessible to readers not specialized in the field with brief definitions or explanations.
Suggestion: When introducing technical terms or specific tests (e.g., OKC, CKC exercises), a short parenthetical explanation could make the discussion more accessible to a broader audience.
For Conclusions:
1. Clearly State the Main Findings:
Start with a clear, concise statement of the main findings, emphasizing the lack of difference in functional outcomes in operated knees and highlighting the unique finding regarding non-operated knees.
Suggestion: "Our study found no significant differences in functional outcomes—knee muscle strength, dynamic postural stability, or self-reported outcomes—between ACLR with meniscal repair for stable versus unstable meniscal tears in operated knees."
2. Emphasize the Novelty of the Findings:
Highlight the uniqueness of the findings regarding non-operated knees, as this seems to be a significant contribution of the study.
Suggestion: "A novel finding of our study is the poorer dynamic postural stability observed at 6 months postoperatively in non-operated knees for patients with unstable meniscal tears, a crucial insight for postoperative rehabilitation."
3. Link Findings to Practical Implications:
Explicitly connect the study's findings to their practical implications for clinical practice, particularly the importance of including balance exercises in rehabilitation protocols.
Suggestion: "Given the observed correlation between preoperative/postoperative stabilities in operated knees and postoperative stability in non-operated knees, our results underscore the importance of integrating balance exercises for both knees in the rehabilitation protocol, particularly for patients with unstable meniscal tears."
4. Recommendations for Future Research:
While the conclusion primarily focuses on clinical recommendations, briefly mentioning future research directions could enrich the conclusion.
Suggestion: "Further research is needed to explore the long-term effects of different rehabilitation protocols on dynamic postural stability in non-operated knees, as well as the potential biomechanical and neurological mechanisms underpinning our findings."
5. Highlight the Clinical Relevance:
Make the clinical relevance of the study's findings explicit, particularly for practitioners designing rehabilitation protocols.
Suggestion: "Our findings have direct clinical implications, suggesting that clinicians should consider the stability of non-operated knees in designing rehabilitation protocols after ACLR, especially for patients with unstable meniscal tears."
6. Cautiously Address the Recommendation:
While recommending balance exercises for both knees is based on the findings, it may be beneficial to acknowledge any limitations that might affect the generalizability of this recommendation.
Suggestion: "While we recommend balance exercises for both knees to enhance postoperative recovery, especially in patients with unstable meniscal tears, it's crucial to tailor these exercises to individual patient needs and consider other factors such as age and overall physical condition."
Comments on the Quality of English LanguageGeneral Suggestions Across All Sections
1. Consistency in Hyphenation and Compound Words:
Ensure consistency in the use of hyphenation for terms like "non-operated" and "postoperative." Decide on one format and apply it throughout the document.
2. Use of Acronyms:
Upon first mention of an acronym, the term should be spelled out with the acronym in parentheses, then the acronym can be used alone thereafter. This seems to be handled well, but it's a common oversight worth checking.
3. Correct Use of Articles:
Pay attention to the use of articles ("a", "an", "the") for general readability and grammatical correctness, especially in complex sentences describing methods and results.
Specific Suggestions
Abstract
"A total of 76 patients... were analyzed": Consider specifying that patients were "randomly selected" if applicable, to clarify the selection process.
"In non-operated knees, there was a significant group-by-time interaction effect": Specify "a significant difference was observed" to clarify what kind of effect.
Introduction
"Insufficient recovery of functional outcomes may also be related to persistent discomfort, reinjury, and low surgery-related satisfaction": Consider rephrasing for clarity, "Insufficient recovery from functional outcomes may lead to persistent discomfort, increased risk of reinjury, and reduced satisfaction with surgery outcomes."
Materials and Methods
"This prospective comparative study was approved by": Specify "The Institutional Review Board (IRB) approved this prospective comparative study" for clarity.
Ensure technical descriptions are clear and concise. For example, when describing the rehabilitation protocol, use bullet points or numbered lists for readability.
Results
Consider adding a summary sentence at the beginning of the Results section for an overview of the findings.
"No statistically significant differences were found": This phrase is clear, but you might add what this suggests about the hypotheses for clarity, e.g., "indicating that...".
Discussion
The discussion mixes past and present tense; maintain consistency, typically using past tense for what was done or found and present tense for its current implications.
"This result might be explained by altered central somatosensory pathways": It could be strengthened by stating, "This suggests that altered central somatosensory pathways may contribute to...".
Conclusions
"Therefore, we suggest that balance exercises should be performed in both knees after surgery": Be assertive with recommendations based on your findings. "Based on our findings, we recommend incorporating balance exercises for both knees in post-surgical rehabilitation, especially..."
Round 2
Reviewer 4 Report
Comments and Suggestions for Authors
Dear Autors,
Following a second review of your manuscript titled “Dynamic postural stability was poorer in non-operated knees after anterior cruciate ligament reconstruction with meniscal repair for unstable meniscal tears than for stable meniscal tears,” I am pleased to confirm that the revisions made thoroughly address my previous comments. The manuscript is now significantly improved, and the responses provided clearly articulate the enhancements and justification for each change made.
As such, I recommend the manuscript for acceptance in its current form.
Best regards